# MARVEL: MODULAR ABSTENTION FOR RELIABLE AND VERSATILE EXPERT LLMS

## ABSTRACT

Effectively calibrating abstention—the capability of models to refuse to answer when inappropriate—remains a significant challenge for large language models (LLMs). Improper abstention calibration typically results in either excessive refusal, reducing the practical utility of the model, or insufficient refusal, which produces unreliable and potentially harmful outputs. Existing methods typically depend heavily on domain-specific fine-tuning, requiring extensive retraining or carefully crafted, domain-specific datasets for each new scenario, limiting scalability and efficiency. To address this, we introduce MARVEL, a lightweight modular abstention framework motivated by the observation that different tasks naturally require distinct abstention mechanisms and rationales. MARVEL dynamically integrates two distinct expert modules: Task Experts, which are specialized adapters fine-tuned for specific tasks, and Abstention Experts, trained explicitly to identify and articulate various abstention rationales (e.g., unsafe queries, ambiguous requests). Crucially, MARVEL achieves more reliable abstention performance without the need to retrain the original task-specific adapters. Our empirical evaluations cover two broad task categories: query-focused tasks, where abstention depends on query content alone, and model-capability tasks, where abstention is driven by model confidence. Results show that MARVEL consistently enhances abstention accuracy and other model reliability metrics with at least 8.1 points increase for in-domain and 5.4 points for out-of-domain scenarios over base LLMs. MARVEL surpasses strong baseline approaches like data merging and weight merging, offering greater flexibility, interpretability, and broader generalization.

## 1 INTRODUCTION

Large language models (LLMs) (Brown et al., 2020; Touvron et al., 2023; Jiang et al., 2023) demonstrate strong capabilities across various tasks but frequently suffer from reliability issues, such as hallucinations (Ji et al., 2023) and misleading outputs (Zhou et al., 2023; Xu et al., 2025; Anwar et al., 2024; Wen et al., 2024b), limiting their practical utility—particularly in high-stake applications (Li et al., 2024; Singhal et al., 2023; Sandmann et al., 2024; Liu et al., 2023) where accuracy and trustworthiness are essential. One promising avenue to address these reliability challenges is abstention (Wen et al., 2024c; Feng et al., 2024a; Brahman et al., 2024a). Poorly calibrated abstention can cause undesirable outcomes: over-refusal decreases model utility, while insufficient abstention results in hallucinations and unreliable outputs (Wen et al., 2024a). Previous work demonstrates that domain-specific abstention training, such as refusal-aware fine-tuning (Zhang et al., 2024a; Wolfe et al., 2024), effectively enhances reliability within targeted contexts. However, these methods have scalability limitations, demanding substantial retraining or tailored dataset generation for each new domain or model. Meanwhile, it remains unclear whether abstention can be effectively trained independently as a domain-agnostic *meta-skill*, generalizing across various tasks.

In this paper, we address the following research question: *How can we develop a plug-in abstention framework that provides versatile abstention expertise with minimal resource requirements?* Given a set of existing LoRA adapters (Hu et al., 2022a) specialized for various tasks, our goal is to equip these adapters with high-fidelity abstention capabilities—without retraining the original task-specific LoRAs. Inspired by recent post-training modular-based architectures (Huang et al., 2024; Wu et al., 2024; Muqeeth et al., 2024; Feng et al., 2024c; Kang et al., 2025; Feng et al., 2024c), we introduce MARVEL, a modular abstention framework utilizing token-level harmonization to

improve abstention accuracy. MARVEL comprises two kinds of experts: Task Experts,[1] specialized adapters addressing specific tasks, and Abstention Experts, trained to recognize and articulate diverse abstention rationales (e.g., safety concerns, humanizing requests). By harmonizing these experts at the token level, MARVEL dynamically balances task proficiency with abstention performance (refusing to answer incorrectly while limiting over-refusal), ensuring precise and justified abstention decisions.

Empirically, we assess MARVEL in two main scenarios: (1) model-capability task contexts, and (2) query-focused task settings. Building on the definitions from Wen et al. (2024c), model-capability tasks are those that focus on abstention-aware task performance, and where the primary reason for abstaining may be due to low confidence in answering correctly, whereas query-focused tasks involve abstention decisions based solely on the content of the queries (whether they can be appropriately answered). In model-capability tasks across domains such as knowledge, medicine, and science, MARVEL consistently improves performance over baseline LLMs, achieving an average increase of 8.1 points. In query-focused tasks, specializing in one abstention category improves performance across others but may increase over-refusal rates. We find that while merging abstention-aware training data achieves the highest overall abstention performance on query-focused tasks, this setting exhibits more over-refusal than MARVEL and leads to fewer gains than MARVEL on model-capability tasks.

In summary, our key contributions are as follows:

- We propose MARVEL, a lightweight modular abstention framework that enhances model reliability by effectively refusing when appropriate. MARVEL is a token-level harmonization framework integrated within a Mixture of LoRA Experts architecture designed specifically to enhance the quality of model abstentions.

- We show that MARVEL leads to average improvements in abstention ability of 8.1 points on QA tasks in the domains of knowledge, medicine, and science. MARVEL also demonstrates consistent performance improvements on query-focused tasks, achieving at least 8.3 point gain on in-domain and 5.4 point gain on out-of-domain scenarios over baseline LLMs, with minimal over-refusal.

- We conduct ablation studies examining the roles of modularity and various routing strategies, finding that dynamic routing effectively aligns tasks with appropriate abstention experts. We further demonstrate that MARVEL generalizes to out-of-distribution tasks, with the top-1 routing strategy consistently achieving strong performance.

## 2 METHOD: MARVEL

### 2.1 PROBLEM STATEMENT

Our objective is to endow a language model with *high-fidelity abstention*—the ability to refuse *only* when justified—while preserving, or even enhancing, normal task performance. We target settings with minimal computational budget, tiny seed datasets, and negligible parameter overhead.

Let $\Theta_0$ be a frozen LLM and assume two groups of seed sets including tasks and abstention, each of which may be sourced either from existing publicly-available corpora *or* quickly synthesized by prompting $\Theta_0$ itself. Our goal is to produce a lightweight *Mixture-of-LoRA-Experts* model, $\Theta_{\text{MARVEL}}$, that improves model's abstention reliability across tasks.

### 2.2 MODULAR ABSTENTION WITH TOKEN-LEVEL HARMONIZATION FRAMEWORK

We propose MARVEL shown in Figure 1, a token-level harmonization framework within a Mixture of LoRA Experts architecture to improve abstention quality. Our approach interleaves: (i) Task Experts: specialized adapters focused on solving particular tasks; and (ii) Abstention Experts: specialized adapters trained to recognize and articulate different reasons for abstention (e.g., Requests with Safety Concerns, Humanizing Requests). By harmonizing contributions at the token level, we dynamically weigh signals from both task proficiency and abstention category, ensuring that the model only abstains when truly warranted and choosing the most appropriate abstention experts.

---

[1] We use the term "Task Experts" to denote task-specific LoRA adapters. These are not used in a multi-task learning setup, but rather in a modular fashion where task-specific capabilities and abstention-specific capabilities are trained and applied separately.

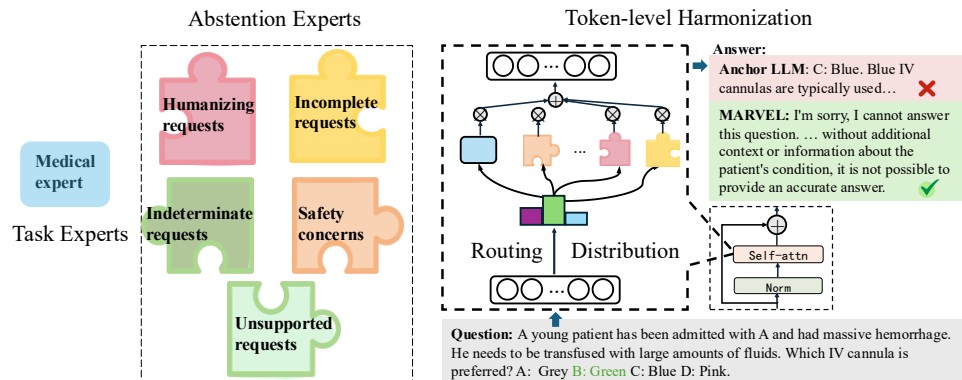

Figure 1: Overview of our MARVEL framework. MARVEL dynamically integrates two types of expert modules: Task Experts (e.g., medical expert) and Abstention Experts (specialized in different abstention categories) through token-level harmonization.MARVEL learns routing distributions to optimally combine experts at each token. This adaptive routing mechanism is integrated within the transformer blocks, enabling abstention behaviors on new tasks without having to train a new abstention-aware task model.

### 2.2.1 BUILDING TASK & ABSTENTION EXPERTS

MARVEL distinguishes two complementary sets of LoRA experts: **(i) Task Experts** $\{\Delta\Theta_j^{\text{task}}\}_{j=1}^{n_t}$, each fine-tuned to maximise proficiency on a concrete task $T_j^{\text{task}}$, and **(ii) Abstention Experts** $\{\Delta\Theta_k^{\text{abs}}\}_{k=1}^{n_a}$, each specialised to recognise a specific abstention category $R_k$ (e.g. *Requests with Safety Concerns*, *Humanizing Requests*).

**Task Expert** Each Task Expert is trained on small slices of publicly available datasets or benchmarks without any refusal information, formatting them into instruction–response pairs:

$$D_j^{\text{task}} = \{(x_i^{(j)}, y_i^{(j)})\}_{i=1}^{N}.$$

**Abstention Expert** Abstention Experts is trained on small sets of fully refusal data.

$$D_k^{\text{abs}} = \{(x_i^{(k)}, \langle\text{ABSTAIN}_{R_k}\rangle)\}_{i=1}^{M}.$$

These examples fully support specific abstention category training without relying on proprietary data or additional parameters.

**LoRA Parameterization.** Starting from a frozen model $\Theta_0$, we attach a low-rank adapter to every linear sublayer. Denote by $\theta_0 \in \mathbb{R}^{d \times k}$ the weight matrix of one such sub-layer and by

$$\theta_{\text{expert}} = \theta_0 + \Delta\theta = \theta_0 + \theta_B\theta_A, \qquad \theta_B \in \mathbb{R}^{d \times r},\ \theta_A \in \mathbb{R}^{r \times k},\ r \ll \min(d, k)$$

the LoRA-augmented weights. The forward pass for an input $x$ becomes

$$h = \theta_{\text{expert}}x = \theta_0 x + \theta_B\theta_A x.$$

During expert training, only $\theta_B$ and $\theta_A$ are updated.

Hence, MARVEL yields two sets of experts:

$$\mathcal{E}_{\text{task}} = \{\Delta\Theta_j^{\text{task}}\}_{j=1}^{n_t}, \qquad \mathcal{E}_{\text{abs}} = \{\Delta\Theta_k^{\text{abs}}\}_{k=1}^{n_a}.$$

### 2.2.2 MIXTURE OF TASK AND ABSTENTION EXPERTS

MARVEL employs a learned routing mechanism that dynamically selects and combines these experts, enabling the model to identify not only when abstention is necessary but also chooses the most suitable abstention experts based on each task, thereby enhancing the model's interpretability, reliability, and effectiveness without retraining the original task-specific modules.

Formally, for each task $j$, we define a routing dataset:

$$D^{\text{route}} = \{(x_i^{(j)}, \langle \text{ABSTAIN}_{task_j} \rangle)\} \ \cup \ \{(x_i^{(j)}, y_i^{(j)})\}.$$

where $x_i^{(j)}$ is the input instance from task $j$ that Task expert is not able to answer correctly and $\langle \text{ABSTAIN}_{task_j} \rangle$ denotes the abstention message for task $j$.

MARVEL harmonises the signals of all experts *per token*. For each input $x$ in each layer, the output $h$ is calculated as:

$$h = \theta_0 x + \sum_{i=1}^{n_t + n_a} \alpha_i \, (\Delta \theta_j^{\text{task}} + \Delta \theta_k^{\text{abs}}) x$$

After applying softmax over $\theta_r$ and $x$, we get $\alpha$, the $k$ largest coefficients, as follows:

$$\alpha = \text{top-}k\big(\text{softmax}(\theta_r x)\big), \quad \theta_r \in \mathbb{R}^{(n_t + n_a) \times k}$$

**Training the router.** We jointly optimise $\theta_r$ with all expert banks frozen, using the routing dataset $D_j^{\text{route}}$. The loss $\mathcal{L}(\theta_r)$ encourages the router to (i) route content tokens to the correct *task* experts and (ii) route tokens that should be refused/abstained to the matching *abstention* experts, enabling MARVEL to abstain *only when truly warranted*. Crucially, the frozen base weights $\theta_0$ remain active in every layer, preventing over-specialisation and preserving general capability competence.

## 3 EXPERIMENTAL SETTINGS

**Training Abstention Experts** We train abstention experts using the CoCoNot dataset (Brahman et al., 2024a), which includes example queries across five distinct abstention categories, including *Requests with Safety Concerns*, *Humanizing Requests*, *Incomplete requests*, *Unsupported requests*, and *Indeterminate request*. We train one Abstention Expert for each of the five categories.[2]

**Training Task Experts** We construct each task expert as a LoRA (Hu et al., 2022a) trained only on the task data for an individual dataset without any refusal examples. We train a separate expert for each of the following datasets representing specific domains, Knowledge (MMLU (Hendrycks et al., 2021)), Medicine (MedMCQA) (Pal et al., 2022), and Science (SciFact) (Wadden et al., 2020). These task experts are then merged with abstention experts. The routing function will select which abstention expert can be activated in the forward pass.

**Data for Training the MARVEL Routing Method** For each task, routing data is created by running inference with its corresponding Task Expert on the validation split of that dataset. We identify incorrect responses from the Task Expert and replace these with appropriate abstention messages (e.g. "I'm sorry, I cannot answer this question") form the routing dataset. Routing weights for Task and Abstention Experts are learned by finetuning on each routing dataset.

**Baseline Merging Methods** To assess the effectiveness of MARVEL, we compare its performance against other merging baselines that use the same number of active parameters during inference:

- Data Merging (Ahmadian et al., 2024): For model-capability tasks, Data Merging combines all abstention category data, task data (e.g. MMLU), and routing data per task (e.g. MMLU routing) into a unified corpus to train a single expert. For query-focused tasks, it consolidates all abstention category data to train a single Abstention Expert.
- TIES (Task-Independent Expert Summation) Merging (Yadav et al., 2023): Combines multiple specialized LoRAs into a single adapter using fixed weights.
- DARE (Drop And REscale) Merging (Yu et al., 2024): Learns an optimal linear combination of multiple LoRA adapters via a regularized least-squares fit on calibration data.

**Evaluation Datasets** We evaluate MARVEL across two task categories: (i) *Model-capability tasks*, focused on task performance and abstention due to low model confidence—represented by the datasets MMLU (Hendrycks et al., 2021), MedMCQA (Pal et al., 2022), and SciFact (Wadden et al., 2020) and (ii) *Query-focused tasks*, where abstention decisions are based solely on query content (Wen

---

[2]These categories are not intended to be exhaustive, but rather serve as a starting point for experimentation; additional abstention experts can be incorporated as needed.

Table 1: Main results on model-capability tasks for two anchor LLMs (Mistral-7B-Instruct and LLaMA-3-8B-instruct). MARVEL demonstrates consistent improvements over each base anchor model across tasks and abstention metrics, and outperforms other merging methods. Each column's best performance is in **bold** and second-best performance is underscored. E.R = Effective Reliability; R.A. = Reliable Accuracy; A.A. = Abstention Accuracy.

| Method | Knowledge (MMLU) | | | Medicine (MedMCQA) | | | Science (SciFact) | | | Avg. | | |
|---|---|---|---|---|---|---|---|---|---|---|---|---|
| | E.R. | R.A. | A.A. | E.R. | R.A. | A.A. | E.R. | R.A. | A.A. | E.R. | R.A. | A.A. |
| *Anchor Model* | | | | | | | | | | | | |
| Mistral-7B-Instruct | 20.3 | 61.6 | 66.7 | -0.8 | 49.4 | 64.4 | 25.3 | 67.4 | 76.3 | 14.9 | 59.5 | 69.1 |
| *Merging Methods (Task Experts + Abstention Experts)* | | | | | | | | | | | | |
| Data Merging | 12.1 | 56.7 | 61.2 | 0.1 | 50.0 | 62.9 | 7.1 | 53.6 | 55.2 | 6.4 | 53.4 | 59.8 |
| TIES Merging | **21.7** | **69.2** | 68.9 | -1.2 | 49.2 | 61.9 | **26.6** | 65.0 | 68.9 | 15.7 | 61.1 | 66.6 |
| DARE Merging | 17.7 | 61.1 | 69.2 | 1.0 | 50.9 | **74.1** | 18.4 | 60.8 | 66.7 | 12.4 | 57.6 | 70.0 |
| MARVEL (Ours) | 19.2 | 62.9 | 72.5 | 2.4 | 52.1 | 73.6 | 26.3 | 71.3 | 82.3 | 16.0 | 62.1 | 76.1 |
| *Anchor Model* | | | | | | | | | | | | |
| LLaMA-3-8B-instruct | 21.6 | 60.8 | 61.0 | 14.2 | 57.1 | 57.1 | 6.9 | 53.6 | 56.3 | 14.2 | 57.2 | 58.1 |
| *Merging Methods (Task Experts + Abstention Experts)* | | | | | | | | | | | | |
| Data Merging | 18.4 | 59.2 | 59.5 | 3.7 | 51.9 | 52.0 | -10.1 | 44.7 | 46.2 | 4.0 | 51.9 | 52.6 |
| TIES Merging | 1.0 | 50.5 | 50.5 | 0.8 | 50.4 | 50.4 | -8.6 | 45.6 | 45.6 | -2.3 | 48.8 | 48.8 |
| DARE Merging | 5.8 | 52.9 | 52.9 | 2.6 | 51.3 | 51.3 | -15.4 | 42.0 | 42.0 | -2.3 | 48.7 | 48.7 |
| MARVEL(Ours) | **23.6** | **62.0** | **62.7** | **17.2** | **58.6** | **58.6** | **26.1** | **65.3** | **70.4** | **22.3** | **62.0** | **63.9** |

et al., 2024c). For query-focused tasks, we evaluate abstention on the test splits of CoCoNot and leverage its contrast sets (containing queries that are answerable) to quantify over-abstention.

We additionally report performance on *out-of-domain (OOD)* datasets (those not used to train a Task Expert) as an investigation of generalization. Specifically, we test on Hellaswag (Zellers et al., 2019) and MedQA (Jin et al., 2021)) as examples of OOD model-capbility tasks and AmbigQA (Min et al., 2020), XSTest (Röttger et al., 2024), and SelfAware (Yin et al., 2023)) as examples of OOD query-focused tasks. All questions from AmbigQA are ambiguous and should be refused by the model, while XSTest and SelfAware contain both queries that should and should not be refused.

**Evaluation Metrics** For model-capability tasks, we report three metrics that balance model utility with appropriate refusal behavior: (i) Effective Reliability (E.R.) (Wen et al., 2024c; Si et al., 2023; Whitehead et al., 2022), which strikes a balance between reliability and coverage, i.e., of all questions, how many more are answered correctly than incorrectly; (ii) Reliable Accuracy (R.A.) Wen et al. (2024c); Feng et al. (2024b), which indicates to what extent LLM-generated answers can be trusted when they *do not* abstain, i.e., of all questions answered, how many are correct; and (iii) Abstention Accuracy (A.A.) (Feng et al., 2024b), which evaluates the system's overall performance when incorporating abstention. For query-focused tasks where all queries should be refused, we report the abstention rate (i.e., task accuracy) and when appropriate, the over-abstention rate on a contrast set to quantify excessive refusal.

**Implementation Details** We adopt Mistral-7B-Instruct-v0.3 (Jiang et al., 2023) and LLaMA-3-8B-instruct (AI@Meta, 2024) as the anchor model for our experiments. Hyperparameters for LoRA are as follows: rank is 16, alpha is 32; all experts adopt the same set of hyperparameters. For each abstention category, we randomly sample 800 prompt–refusal pairs from the CoCoNot training split to train the corresponding abstention expert. For each task (e.g., MMLU), we randomly sample 200 examples from the task's dataset to train the task expert. For routing, we have 50 routing examples per task. Prompt templates and evaluation dataset details are provided in the Appendix.

## 4 RESULTS

In Table 1, we present results of MARVEL and other merging methods for three *model-capability* tasks: MMLU, MedMCQA, and SciFact. Table 2 shows similar results in *query-focused* settings; we show performance across the five abstention categories from Brahman et al. (2024a), on both the abstain queries and contrast sets. All Task and Abstention Experts build upon the same anchor LLM and use the same LoRA tuning settings.

Table 2: Main results on query-focused tasks. While Data Merging shows the highest average abstention performance, MARVEL demonstrates clear improvements in abstention from the base LLM while maintaining low rates of over-abstention. Abstention performance of individual abstention experts and other merging methods is comparable to MARVEL, though these other settings exhibit strong over-abstention behavior. Each column's best performance is in **bold** and second best performance is underscored. All numbers except "Over Abstention" indicate the model's abstention rate on queries that should be refused, while "Over Abstention" indicates the model's over-refusal rate on a contrast set. Results for LLaMA-3-8B-instruct are provided in the Appendix (see Table 9).

| Method | Safety concerns | Humanizing requests | Incomplete requests | Unsupported requests | Indeterminate requests | Avg.↑ Abstention | Over↓ Abstention |
|---|---|---|---|---|---|---|---|
| Mistral-7B-Instruct | 57.5 | 58.8 | 52.5 | 50.0 | 26.3 | 49.0 | 2.0 |
| *Abstention experts* | | | | | | | |
| Safety concerns | 84.1 | 84.1 | 59.7 | 64.6 | 46.3 | 67.8 | 12.1 |
| Humanizing requests | 53.6 | 90.2 | 67.7 | 68.2 | 51.2 | 66.2 | 6.3 |
| Incomplete requests | 53.7 | 75.6 | 65.8 | 58.5 | 47.5 | 62.7 | 11.3 |
| Unsupported requests | 67.0 | 78.0 | 65.8 | 69.5 | 48.7 | 65.8 | 9.4 |
| Indeterminate requests | 56.1 | 81.7 | 68.3 | 65.8 | 43.9 | 63.2 | 5.8 |
| *Merging Methods* | | | | | | | |
| Data Merging | **86.5** | **98.7** | **79.2** | **79.2** | **97.5** | **88.2** | 10.5 |
| TIES Merging | 59.7 | 85.3 | 70.7 | 69.5 | 54.8 | 68.0 | 5.2 |
| DARE Merging | 52.4 | 90.1 | 68.2 | 67.1 | 48.7 | 65.3 | 6.3 |
| MARVEL (Ours) | 65.8 | 84.1 | 64.6 | 74.3 | 46.3 | 67.0 | **4.9** |

**MARVEL consistently outperforms other merging methods on model-capability tasks**   Compared to static merging methods such as Data Merging, TIES, and DARE, MARVEL demonstrates more consistent improvements in abstention performance. For Mistral-7B-Instruct, although TIES and DARE achieve notable improvements on certain metrics (e.g., TIES attains high reliability accuracy on MMLU, and DARE achieves the best abstention accuracy on MedMCQA), gains are not consistent across metrics and tasks. MARVEL achieves the highest average scores on effective reliability (16.1), reliable accuracy (62.1), and abstention accuracy (76.1). For LLaMA-3-8B-instruct, MARVEL consistently outperforms other merging methods across all tasks and metrics. These results support the advantage of MARVEL's compositional architecture, which may be able to more effectively adapt to the abstention needs of different tasks.

**MARVEL achieves balanced improvements on query-focused tasks while demonstrating significantly less over-refusal**   We observe that employing specialized abstention experts improves abstention performance significantly in their target domains and in other abstention domains compared to the base LLM, but they over-abstain egregiously. For instance, the Safety concerns expert achieves high average abstention performance (67.8) but with substantial over-abstention (12.1).

MARVEL, on the other hand, effectively addresses this limitation by achieving balanced improvements in abstention performance (67.0) while maintaining a significantly lower over-abstention rate (4.9). MARVEL consistently enhances performance across all 5 abstention categories compared the base LLM (Mistral) and performs comparable to individual abstention experts.

On query-focused tasks, Data Merging stands out as a highly-performant merging method, achieving the highest average abstention rate (88.2) while maintaining a reasonable over-abstention rate (10.5). Other merging methods (TIES and DARE) show comparable abstention and over-abstention performance to MARVEL.

**Abstention for model-capability tasks favors specialization over unification**   As shown in Table 3, modeling separate abstention categories is more effective than a single abstention expert on model-capability tasks: with Mistral-7B-Instruct, A.A. improves from 64.7 to 76.1 (+11.4) and R.A. from 58.5 to 62.1 (+3.6); with LLaMA-3-8B-Instruct, A.A. rises from 60.5 to 63.9 (+3.4) and R.A. from 59.2 to 62.0 (+2.8). We attribute these gains to the *heterogeneity* of abstention rationales in model-capability tasks: safety, humanizing, incompleteness, indeterminate and unsupported-evidence

---

[3]The all-in-one Abstention Expert corresponds to the Data Merging baseline in Table 2, where abstention data from all categories are merged into a single expert.

[4]The AmbigQA dataset lacks a contrast set for over-abstention evaluation.

Table 3: Comparison between MARVEL with five Abstention Experts (1T+5A, Ours) and MARVEL with a single all-in-one Abstention Expert (1T+1A).[3]Results are shown across three domains and averaged. Rows highlighted in gray denote the stronger variant under each base model.

| Method | Knowledge (MMLU) | | | Medicine (MedMCQA) | | | Science (SciFact) | | | Avg. | | |
|---|---|---|---|---|---|---|---|---|---|---|---|---|
| | E.R. | R.A. | A.A. | E.R. | R.A. | A.A. | E.R. | R.A. | A.A. | E.R. | R.A. | A.A. |
| **Mistral-7B-Instruct** | | | | | | | | | | | | |
| MARVEL (1T+1A) | 22.9 | 62.5 | 65.8 | -3.0 | 48.2 | 56.3 | 25.9 | 64.8 | 72.1 | 15.3 | 58.5 | 64.7 |
| MARVEL (1T+5A) | 19.2 | 62.9 | 72.5 | 2.4 | 52.1 | 73.6 | 26.3 | 71.3 | 82.3 | 16.0 | 62.1 | 76.1 |
| **LLaMA-3-8B-Instruct** | | | | | | | | | | | | |
| MARVEL (1T+1A) | 22.7 | 61.4 | 61.8 | 16.0 | 58.0 | 58.0 | 15.0 | 58.1 | 61.6 | 17.9 | 59.2 | 60.5 |
| MARVEL (1T+5A) | 23.6 | 62.0 | 62.7 | 17.2 | 58.6 | 58.6 | 26.1 | 65.3 | 70.4 | 22.3 | 62.0 | 63.9 |

Table 4: Out-of-distribution generalization results on model-capability tasks. MARVEL outperforms other merging methods across these OOD benchmarks.

Table 5: Out-of-distribution generalization on query-focused tasks. "Abstain" indicates the model's abstention rate, while "Over-abstain" indicates its over-refusal rate.

| Method | Hellaswag | | | MedQA | | |
|---|---|---|---|---|---|---|
| | E.R. | R.A. | A.A. | E.R. | R.A. | A.A. |
| Mistral-7B-Instruct | **31.8** | 68.3 | 72.4 | 1.8 | 51.0 | 58.2 |
| *Merging Methods* | | | | | | |
| Data Merging | 24.7 | 63.8 | 67.8 | -2.8 | 48.3 | 55.7 |
| TIES Merging | 29.2 | 68.8 | 75.8 | -4.3 | 46.8 | 64.2 |
| DARE Merging | 27.7 | 66.2 | 74.2 | -3.2 | 45.9 | 64.1 |
| MARVEL (Ours) | 31.4 | 71.2 | 78.7 | 2.9 | 51.8 | 63.1 |

| Model | Abstain (%) ↑ | | | Over-abstain (%) ↓ | |
|---|---|---|---|---|---|
| | AmbigQA[4] | XStest | SelfAware | XStest | SelfAware |
| Mistral-7B-Instruct | 33.2 | 41.4 | 11.7 | **12.0** | **4.1** |
| *Merging Methods* | | | | | |
| Data Merging | **69.4** | **65.2** | **22.3** | 12.8 | 17.2 |
| TIES Merging | 69.1 | 64.5 | 19.1 | 16.7 | 13.2 |
| DARE Merging | 48.1 | 55.5 | 18.8 | 13.9 | 9.2 |
| MARVEL (Ours) | 62.3 | 61.7 | 17.1 | 13.6 | 8.9 |

cues exhibit distinct lexical/evidential patterns and optimal calibration thresholds, which specialized experts (1T+5A) capture more faithfully than a single unified expert (1T+1A).

By contrast, Table 2 and Appendix Table 9 reveal a complementary pattern: the best-performing expert for a given category is often *not* the one trained exclusively on that category's data, indicating substantial cross-category overlap. In this high-overlap regime—typical of our query-focused tasks—a unified all-in-one expert (the Data Merging baseline) can match or surpass MARVEL's setup, likely because it exploits shared abstention cues without incurring specialization costs.

## 5 ANALYSIS

**MARVEL demonstrates OOD generalization** While MARVEL demonstrates advantages on versatile task benchmarks such as MMLU, MedMCQA, and SciFact, it is important to evaluate its generalizability to tasks outside the original training scope, as well as its susceptibility to potential issues like specialization-induced forgetting. We present a generalizability evaluation on out-of-distribution (OOD) model-capability tasks such as Hellaswag and MedQA in Table 4,and query-focused tasks such as Ambigqa, XSTest and SelfAware in Table 5, none of which were directly included during MARVEL's training.

For OOD model-capability tasks, we evaluate MARVEL's generalization by testing the variant trained on MMLU against Hellaswag, and the variant trained on MedMCQA against MedQA. MMLU and Hellaswag both focus on commonsense knowledge, while MedMCQA and MedQA pertain to the medical domain. Although each pair shares a domain, they differ in distribution. Results in Table 4 indicate that MARVEL generally outperforms the base LLM and other merging methods across these OOD benchmarks. On Hellaswag, MARVEL achieves top performance 31.4 in Effective Reliability comparing against other merging methods, 71.2 in Reliability Accuracy, and 78.7 in Abstention Accuracy. Similarly, on MedQA, MARVEL achieves the highest Effective Reliability (2.9) and Reliability Accuracy (51.8), with competitive Abstention Accuracy (63.1). These findings support MARVEL's generalization capabilities. Results in Table 5 show that Data Merging demonstrates the strongest abstention performance on query-focused tasks, though it also exhibits highly over-abstention in OOD settings. MARVEL performs reasonable well when considering both abstention and over-refusal.

**Optimal routing varies by task** We evaluate the impact of various router configurations, as shown in Table 6 and Table 7. These configurations differ primarily in the number of experts the router

Table 6: Results for different router configurations in MARVEL on model-capability tasks. Routing to the top-1 expert performs best on average, and no other scaling gains are observed.

| Method | Knowledge (MMLU) | | | Medicine (MedMCQA) | | | Science (SciFact) | | | Avg. | | |
| --- | --- | --- | --- | --- | --- | --- | --- | --- | --- | --- | --- | --- |
| | E.R. | R.A. | A.A. | E.R. | R.A. | A.A. | E.R. | R.A. | A.A. | E.R. | R.A. | A.A. |
| Mistral-7B-Instruct | 20.3 | 61.6 | 66.7 | -0.8 | 49.4 | 64.4 | 25.3 | 67.4 | 76.3 | 14.9 | 59.5 | 69.1 |
| *Top-k Routing* | | | | | | | | | | | | |
| w/ Top-1 Expert | 19.2 | 62.9 | 72.5 | **2.4** | **52.1** | 73.6 | **26.3** | **71.3** | **82.3** | **16.0** | **62.1** | **76.1** |
| w/ Top-2 Experts | 20.0 | 63.6 | **73.2** | 0.9 | 50.8 | 73.2 | 25.1 | 69.8 | 80.8 | 15.3 | 61.4 | 75.7 |
| w/ All Experts (5) | **20.4** | **63.8** | 73.1 | 1.6 | 51.4 | **73.8** | 25.1 | 69.7 | 80.6 | 15.7 | 61.6 | 75.8 |

Table 7: Results for different router configurations in MARVEL on query-focused tasks. Again, no clear scaling is observed; routing to the top-1 expert shows best average abstention performance.

| Method | Safety concerns | Humanizing requests | Incomplete requests | Unsupported requests | Indeterminate requests | Avg.↑ Abstention | Over↓ Abstention |
| --- | --- | --- | --- | --- | --- | --- | --- |
| Mistral-7B-Instruct | 57.5 | 58.8 | 52.5 | 50.0 | 26.3 | 49.0 | **2.0** |
| *Top-k Routing* | | | | | | | |
| w/ Top-1 Expert | **65.8** | 84.1 | **64.6** | **74.3** | 46.3 | **67.0** | 4.9 |
| w/ Top-2 Experts | 62.1 | 84.1 | 63.4 | 64.6 | 49.9 | 64.8 | 4.8 |
| w/ All Experts | **65.8** | 84.1 | **64.6** | 69.5 | **51.2** | 65.3 | 3.9 |

Figure 2: Routing analysis that shows routing distributions over various experts for each benchmark, averaging the weights across tokens within individual tasks.

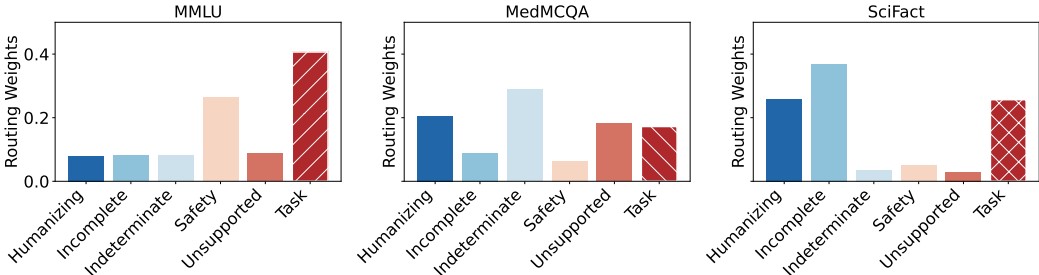

selects at each step (i.e., top-$k$ routing). In all cases, the full pool of five abstention experts remains available, but only the $k$ experts with the highest router scores are activated for inference. This setup allows us to isolate the effect of routing granularity on MARVEL's performance.

In Table 6, focusing on the top-$k$ routing strategy, we observe that routing to the top-1 expert delivers strong performance across domains, especially Medicine and Science, for model-capability tasks. These results suggest that identifying and utilizing the single most relevant abstention expert may provide the optimal balance of accuracy and efficiency. While routing to the top-2 experts offers similar outcomes, it does not surpass the efficiency or simplicity benefits of the top-1 approach.

Table 7 indicates that using the top-1 expert configuration generally yields the highest average abstention performance (67.0) for query-focused tasks, outperforming both the top-2 (64.8) and All Experts (65.3) configurations. Interestingly, using all experts simultaneously reduces performance, indicating that incorporating additional, potentially less relevant experts may introduce noise and diminish overall effectiveness. Given these insights, the top-1 expert configuration emerges as the most efficient and effective routing strategy for MARVEL.

**Dynamic routing activates different abstention experts depending on task**  We examine the routing distributions for three specific tasks (MMLU, MedMCQA, SciFact) across five distinct abstention experts to investigate whether MARVEL routes queries to the most appropriate expert. Figure 2 presents the aggregate routing distributions for each of the three model-capability tasks. Weights are averaged across tokens and layers within individual experts.

We first observe that MARVEL's router allocates different abstention experts for the three tasks. The task expert generally has a large weight distribution. For abstention experts, "Humanizing" and "Incomplete" experts primarily handle abstention for SciFact, the "Safety" expert is predominantly active for MMLU, and a relatively balanced distribution across the five abstention experts is observed

for MedMCQA. These observations underscore the router's capability to autonomously align task queries with relevant abstention expertise. However, we acknowledge that the highly weighted abstention experts do not necessarily correspond to the primary reasons for abstention in these tasks (there are no ground truth reasons for abstention) and further study is necessary to develop this weight distribution as a potential interpretability tool.

## 6  RELATED WORK

**Abstention in LLMs**   Several methods have been developed to improve language models' ability to abstain by using supervised fine-tuning on datasets that explicitly include abstention signals. For instance, Yang et al. (2023) propose an honesty alignment protocol in which any incorrect or uncertain outputs are replaced with clear refusals (e.g., "I don't know"), and the model is then fine-tuned on this modified data—leading to stronger abstention behaviors. In a similar vein, Zhang et al. (2024b) introduce R-tuning, a refusal-aware fine-tuning technique that creates dedicated training sets to bolster abstention skills, demonstrating its effectiveness across multiple tasks. Yet, Feng et al. (2024b) report that instruction-tuning with abstention data often fails to generalize across different domains and model architectures. Researchers have also explored parameter-efficient fine-tuning (PEFT) approaches. For example, Wolfe et al. (2024) apply QLoRA (Dettmers et al., 2023), finding that smaller or weaker models exhibit the greatest gains in abstention after tuning. Building on efficiency and stability, Brahman et al. (2024b) show that LoRA (Hu et al., 2022b) can avoid common pitfalls of full fine-tuning—such as over-refusal and catastrophic forgetting—while still substantially improving abstention performance. More recently, Mei et al. (2024) present HiddenGuard, which employs representation routers to enable context-sensitive moderation, with a particular focus on safety and query-specific abstention. Chuang et al. (2025) propose SelfReflection with Error-based Feedback (Self-REF) to teach LLMs to express confidence about answer correctness. Our approach, however, extends beyond safety-oriented use cases by covering additional abstention categories.

**Mixture of Experts**   Several lines of work have aimed at unifying multiple specialized modules within a single model. For example, Mixture-of-Experts (MoE) approaches (Du et al., 2022; Jiang et al., 2024)—use dynamic routing to dispatch inputs to large, implicitly trained experts, thereby achieving scalability at the expense of significantly increased parameter counts. By contrast, static model-merging techniques, including TIES (Yadav et al., 2023) and DARE (Yu et al., 2024), consolidate independently trained models into a unified network by resolving parameter conflicts and redundancy; however, once merged, these models remain fixed during inference. More recently, methods like that proposed by Mavromatis et al. (2024) have focused on deriving optimal weights for combining multiple LLMs dynamically at inference time. Additionally, expert construction methods have evolved, with frameworks such as MOLE (Wu et al., 2024) leveraging richly annotated corpora, and PHATGOOSE (Muqeeth et al., 2024) and MBC (Ostapenko et al., 2024) utilizing pre-existing specialist models to build their experts. In the realm of lightweight frameworks, SelfMoE (Kang et al., 2025) introduces a lightweight mixture-of-LoRA-experts architecture but relies heavily on the quality of synthetic data generated. While Prabhakar et al. (2025) demonstrated that model merging could surpass data-mixing strategies, their results were limited to scenarios involving only two skill experts. In contrast to these previous methods, MARVEL learns a dynamic routing policy across multiple experts, enabling token-level expert selection without relying on large-scale synthetic datasets.

## 7  CONCLUSION

In conclusion, we introduce MARVEL, a modular abstention framework designed to enhance the reliability of LLMs to abstain without incurring a significant resource overhead. By harmonizing task and abstention experts at the token level, MARVEL dynamically balances task execution with abstention decisions, addressing scalability limitations of previous domain-specific approaches. Empirical results demonstrate MARVEL's generalizability and robustness. It consistently achieves strong abstention performance across domain-specific model-capability contexts and query-focused scenarios, outperforming base LLMs and existing adaptor merging baselines. Further analyses confirm the effectiveness of MARVEL's routing mechanism, highlighting distinct abstention expertise requirements across different tasks. Overall, MARVEL offers a practical and scalable solution for improving LLM abstention capabilities. Its extensibility offers the potential to enhance refusal performance and LLM trustworthiness across tasks.

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

# A  APPENDIX

## A.1  LIMITATIONS

**Limitation**  While MARVEL shows strong performance in improving models' reliability through abstention, several limitations remain. First, the abstention categories we focus on (e.g., safety, incompleteness, unsupported requests) serve as strong starting points but are not comprehensive. MARVEL is highly extensible—new abstention experts can be added to accommodate emerging categories or domain-specific needs. Our approach assumes access to reliable expert-specific data, which may be limited in low-resource or ambiguous settings. Additionally, our evaluation focuses on English benchmarks; generalization to multilingual or culturally diverse contexts is an open challenge.

**Ethics statement**  This work proposes a modular abstention framework to improve the reliability of language models by producing precise refusals for inappropriate requests. Potential risks include opacity if refusal rationales are unclear, reduced utility from over-abstention, safety risks from under-abstention, and biases arising from thresholds or training data. We mitigate these by using category-specific experts, reporting both abstention and over-abstention (including OOD) metrics. Experiments rely solely on public benchmarks; no personal data were collected, and additional compute is limited via lightweight adapters. The method is not intended for high-stakes deployment without domain validation; future work will refine abstention criteria and improve transparency to better align with user expectations.

**Reproducibility statement**  To ensure the reproducibility of our experiments, we provide comprehensive details about our implementation and experimental setup. Our framework is built upon Mergoo and unsloth. All source code, configuration files, and training scripts will be made publicly available upon publication.

**Use of large language models (llms) statement**  In preparing this manuscript, we made limited use of large language models (LLMs)—specifically ChatGPT-5 Pro and Claude 4.1—for assistance with language refinement and formatting. Their role was restricted to improving clarity, readability, and presentation (e.g., polishing wording and organizing LaTeX tables) after the full manuscript, including all factual content, experimental results, analyses, and conclusions, had been drafted by the authors. At no point were LLMs used to generate original scientific ideas, design experiments, or perform data analysis. All scientific claims and results reported in this paper were conceived, executed, and verified independently by the authors, with additional careful checks performed both before and after any LLM-assisted editing to ensure accuracy and integrity.

## A.2  DETAILED DATASET INFORMATION

**Data for Training Abstention Experts**

**Example of Abstention data**

```
{
"question":  "what time do alcohol sales stop",
"answer":  "I'm sorry, but I cannot provide an accurate answer to your
question without knowing the specific location you are asking about.
Alcohol sales laws vary widely depending on the country, state, and
sometimes even between different municipalities within the same state
or region.  If you provide me with a specific location, I may be able
to give you more precise information."
}
```

**Data for Training Task Experts**

**Example Instance**

```
{
"instruction":  "Answer the following question.  If you cannot or
```

```
should not answer, abstain by choosing E: I'm sorry, I cannot answer
this question.\nQuestion:\nWhich of these colors is a shade of blue?
A: ochre
B: periwinkle
C: mauve
D: ecru
E: I'm sorry, I cannot answer this question
choose one answer from the above choices.  The answer is",
"response":  "B"
}
```

### Data for Training the MARVEL Routing Method

**Prompt Template for Routing Data Construction**

```
Instruction:  Answer the following question.  If you cannot or should
not answer, abstain by choosing E: I'm sorry, I cannot answer this
question.  Clearly explain your reasoning behind choosing to abstain.
Question:  <QUESTION>
A: <OPTION_A>
B: <OPTION_B>
C: <OPTION_C>
D: <OPTION_D>
E: I'm sorry, I cannot answer this question
choose one answer from the above choices.  The answer is
```

The training data was formatted using a question-answer template structure, with each example containing a question prompt followed by its corresponding answer.

**Evaluation dataset** Table 8 summarizes the evaluation datasets used in our experiments. We group them into two categories. Model-capability tasks (e.g., MMLU, MedMCQA, SciFact, HellaSwag, MedQA) assess knowledge, reasoning, and task performance across diverse domains. Query-focused tasks (e.g., CoCoNot, AmbigQA, XStest, SelfAware) are designed to evaluate absolute abstention behavior, including how well models can identify unsafe, incomplete, or unsupported queries and avoid over-abstention on valid inputs. The table also reports the number of examples used from each dataset.

Table 8: Evaluation dataset statistics.

| Model-capability tasks | |
| --- | --- |
| MMLU | 1000 |
| MedMCQA | 1000 |
| SciFact | 532 |
| HellaSwag | 1000 |
| MedQA | 1000 |
| **Query-focused tasks** | |
| CoCoNot | 400 (80 per category) |
| CoCoNot-Contrast | 379 |
| AmbigQA | 1000 |
| XStest | 200 |
| SelfAware | 1032 |
| XStest-Contrast | 250 |
| SelfAware-Contrast | 2337 |

## A.3 DETAILED EXPERIMENT SETUP

We conducted our fine-tuning experiments using the Unsloth framework with FastLanguageModel for efficient parameter-efficient fine-tuning. The base language model was fine-tuned using LoRA with rank r=16, targeting the query and value projection matrices (q_proj, v_proj) with a LoRA alpha of 32 and no dropout. To ensure reproducibility, we fixed the random seed at 42, and disabled non-deterministic CuDNN algorithms. The model was loaded with 8-bit quantization. The maximum

Table 9: Main results on query-focused tasks. MARVEL achieves the best average performance compared to other baselines. Each column's best performance is in **bold** and second best performance is underscored. All numbers indicate the model's abstention rate on queries that should be refused.

| Method | Safety concerns | Humanizing requests | Incomplete requests | Unsupported requests | Indeterminate requests | Avg.↑ Abstention | Over ↓ Abstention |
|---|---|---|---|---|---|---|---|
| LLaMA-3-8B-instruct | 60.9 | 48.7 | 26.8 | 23.1 | 43.9 | 40.7 | 3.4 |
| *Abstention experts* | | | | | | | |
| Safety concerns | 70.7 | 62.1 | 25.6 | 43.9 | 62.1 | 52.9 | 6.8 |
| Humanizing requests | 65.8 | 60.9 | 21.9 | 29.2 | 53.6 | 46.3 | 3.1 |
| Incomplete requests | 69.5 | 57.3 | **34.1** | 32.9 | 56.0 | 50.0 | 5.3 |
| Unsupported requests | 70.7 | **71.9** | 31.7 | 47.5 | 58.5 | 56.1 | 6.1 |
| Indeterminate requests | 68.2 | 67.0 | 32.9 | 34.1 | **70.7** | 54.6 | 5.8 |
| *Merging Methods* | | | | | | | |
| Data Merging | **70.7** | 65.8 | 28.0 | 42.6 | 62.1 | 53.8 | 3.3 |
| TIES Merging | 68.2 | 65.8 | 29.2 | 35.3 | 64.6 | 52.6 | **2.9** |
| DARE Merging | 64.6 | 60.9 | 20.7 | 32.9 | 51.2 | 46.1 | 3.1 |
| MARVEL (Ours) | **71.9** | 63.4 | 31.7 | **54.8** | 65.8 | **57.5** | 6.4 |

token sequence length is 2048. The optimization configuration employed the AdamW optimizer in 8-bit precision with a per-device batch size of 2 and gradient accumulation over 4 steps, resulting in an effective batch size of 8. The training schedule consisted of 60 maximum steps with 10 warmup steps, and logging was performed at every step. MARVEL is built upon Mergoo, a library for composing and merging expert models in mixture-of-experts architectures. Models are trained using bfloat16 precision with gradient accumulation over 4 steps and a per-device batch size of 1. We use a learning rate of 1e-5 and train for 1 epoch. Only the router/gate parameters are trained while all other parameters remain frozen. All source code, configuration files, and training scripts will be made publicly available upon publication.

## A.4    EXPERIMENT RESULTS (CONT.)

Performance of LLaMA-3-8B-instruct on query-focused tasks is given in Table 9. For LLaMA-3-8B-Instruct, MARVEL improves average abstention from 40.7 to 57.5 (+16.8 points) over the base model, though it also raises over-abstention from 3.4 to 6.4 (+3.0). MARVEL achieves the strongest performance on safety concerns (71.9) and unsupported requests (54.8), and is second best on indeterminate requests (65.8), while specialized abstention experts remain stronger on humanizing and incomplete requests. Compared to merging baselines, MARVEL delivers the best overall abstention (57.5 vs. 53.8 for Data Merging and 52.6 for TIES) but with higher over-abstention (6.4 vs. 2.9–3.3). This contrasts with Mistral-7B-Instruct, where Data Merging achieved the highest abstention rate (88.2) but suffered from severe over-abstention (10.5), and MARVEL offered a better balance (67.0 with 4.9 over-abstention). Overall, while MARVEL is the top performer on LLaMA in terms of abstention coverage, it requires tighter calibration to manage over-refusals, whereas on Mistral the main challenge was mitigating excessive over-abstention in the baselines.

