# OpenReview forum: "MARVEL: Modular Abstention for Reliable and Versatile Expert LLMs"
_ICLR.cc/2026/Conference — ICLR 2026 Conference Withdrawn Submission_

### Official Review · Reviewer_aKu7 · 2025-10-18

**Soundness:** 3
**Presentation:** 3
**Contribution:** 2
**Rating:** 4
**Confidence:** 3

**Summary:**

This paper introduces **MARVEL**, a lightweight and modular abstention framework for large language models (LLMs). MARVEL employs a *Mixture-of-LoRA-Experts* architecture that enables the model to abstain from generating responses when a query is inappropriate or when the model’s confidence is low. Experimental results indicate that MARVEL improves abstention accuracy and model reliability in both in-domain and out-of-domain scenarios.

**Strengths:**

1. The paper is clearly written and easy to follow, with a coherent presentation of the key ideas.
2. The authors conduct detailed analyses and ablation studies that help clarify the contribution of individual components within MARVEL.

**Weaknesses:**

1. **Reproducibility:** The code is not publicly available during the review process, which limits reproducibility and verifiability of the results.
2. **Performance limitations:** As shown in Tables 2 and 5, the proposed method underperforms simple data-merging baselines on out-of-distribution, query-focused tasks. This suggests that MARVEL’s capacity could benefit from further refinement.
3. **Computational overhead:** Since Mixture-of-LoRA-Experts (MoLE) modules cannot be merged back into the main LLM parameters, the framework may introduce additional computational and latency overhead at inference time.
4. **Novelty concerns:** The Mixture-of-LoRA-Experts architecture has previously been used by *SelfMoE* [1], as acknowledged in the related work section. The paper would benefit from a clearer explanation of how MARVEL distinguishes itself from SelfMoE—conceptually, methodologically, or empirically. If MARVEL primarily applies an existing tuning paradigm to a new abstention task, the overall novelty may be limited.

**Minor Concerns:**

1. Figure 1 is conceptually clear but could be visually improved for better readability and aesthetic appeal.
2. The variable *k* is overloaded—representing parameter dimension in line 138 and number of experts in line 175—which may cause confusion. Clarifying this notation would enhance clarity.

> **References**
>
> [1] Self-moe: Towards compositional large language models with self-specialized experts.

**Questions:**

1. How much additional inference latency does MARVEL introduce compared to the base model?
2. In Table 3, using more LoRA experts (e.g., comparing 1T+1A vs. 1T+5A) increases both parameter count and computation. Could the authors comment on whether this comparison is fair, or how computational cost was normalized?
3. Could the authors elaborate on why MARVEL performs differently across model-capability versus query-focused tasks? What underlying factors might explain this variation?
4. Do the LoRA experts specialize as intended—that is, do they activate selectively on tokens or prompts corresponding to their designated categories? A case study or visualization would strengthen this claim.
5. The related work section mentions that SelfMoE relies heavily on the quality of synthetic data. Does MARVEL share this dependency, or is it more robust to data quality variations? If the latter, please explain why.

---

### Official Review · Reviewer_myzx · 2025-10-27

**Soundness:** 2
**Presentation:** 3
**Contribution:** 2
**Rating:** 4
**Confidence:** 3

**Summary:**

This paper introduces MARVEL, a lightweight, modular abstention framework designed to address the challenge of poor abstention calibration in LLMs, which often requires non-scalable, domain-specific fine-tuning. MARVEL dynamically harmonizes two distinct expert modules at the token level: specialized Task Experts and Abstention Experts trained to identify refusal rationales (e.g., unsafe queries, model uncertainty). This approach allows the model to dynamically balance task execution with abstention decisions without needing to retrain the original task adapters. Empirical evaluations on both query-focused and model-capability tasks show that MARVEL significantly improves abstention accuracy and reliability, outperforming base LLMs and other merging baselines, offering a more scalable, generalizable, and practical solution for improving LLM trustworthiness.

**Strengths:**

This paper has the following strengths:

- It proposes MARVEL, a novel, lightweight, and modular abstention framework that integrates a Mixture of LoRA Experts for token-level harmonization, offering a scalable solution to improve abstention quality.

- The proposed method demonstrates substantial, quantified improvements in abstention performance, achieving at least 8.1 point gains on in-domain QA tasks and 5.4 points on out-of-domain scenarios, all while maintaining minimal over-refusal.

- The authors conduct thorough ablation studies that validate the framework's core design choices (modularity and dynamic routing) and successfully demonstrate MARVEL's ability to generalize effectively to out-of-distribution tasks.

**Weaknesses:**

This paper has the following weaknesses:

- The empirical validation is limited to two LLMs (Mistral-7B, Llama-3-8B). The framework's claims of generalizability would be substantially strengthened by evaluating its effectiveness on more recent or architecturally diverse models (e.g., Qwen3 8B and Gemma 3 4b).

- The results tables (e.g., Table 1) report point estimates without measures of variance (like standard deviation) from multiple training runs. Including this is necessary to account for training stochasticity and formally establish the statistical significance of the improvements over baselines.

- The performance gains of MARVEL over merging baselines on query-focused tasks appear marginal. It would be improved by a deeper analysis of these specific results to better understand the framework's limitations or the conditions under which baselines are comparably effective.

**Questions:**

- Can the authors comment on how MARVEL's performance is expected to scale when applied to more recent or architecturally different models (e.g., Qwen3 8B and Gemma 3 4b)?
- To confirm statistical significance against baselines, can the authors provide the variance (e.g., standard deviation across multiple seeds) for the primary results reported in Table 1, accounting for the stochasticity of training?

- What accounts for the marginal performance gains of MARVEL over the merging baselines on query-focused tasks? A deeper analysis of these specific scenarios or potential failure modes would be beneficial.

---

### Official Review · Reviewer_WRCY · 2025-10-29

**Soundness:** 2
**Presentation:** 2
**Contribution:** 2
**Rating:** 2
**Confidence:** 5

**Summary:**

The paper introduces MARVEL, a framework that augments a frozen LLM with LoRA-based task experts and abstention experts, coordinated through a learned token-level router. This design allows the model to respond when confident and abstain when uncertain. Experiments on Mistral-7B and LLaMA3-8B demonstrate improvements over data- and model-merging baselines in both reliability and abstention accuracy.

**Strengths:**

The mixture-of-LoRA-experts architecture with a learned router is conceptually clean and modular, making training straightforward. The paper’s distinction between content-based rejection and confidence-based abstention is well-motivated and aligns with practical deployment scenarios. Also, the transparent presentation of training details supports reproducibility.

**Weaknesses:**

- The abstention signal is artificially constructed with a hindsight of correctness rather than self-consistent. Router training labels are generated by flipping incorrect task answers into abstentions, meaning the router does not learn true epistemic uncertainty but rather correlates abstention with known wrong-answer patterns. This setup risks conflating epistemic uncertainty with empirical error. Given the small training size, lack of theoretical or empirical analysis and high similarity between training/eval data, it remains unclear whether the model learns to abstain when genuinely uncertain or merely memorizes dataset-specific patterns, raising question about its usefulness in real scenario.

- Abstention is conceptually a sentence- or context-level decision, yet the router operates token-by-token. This raises concerns about potential switching between task and abstention experts within the same response, which could yield incoherent or fragmented outputs. The paper does not measure intra-sequence consistency, and since the evaluated datasets feature short QA/MCQ-style answers, these issues remain untested.

- Pattern-based abstention learning could interfere with reasoning or multi-step generation behaviors (e.g., CoT, thinking models like Qwen). It is unclear whether the router’s decision process generalizes beyond memorized lexical or structural cues. Moreover, the paper does not analyze implications for practical inference setups such as majority voting or pass@k aggregation, which are increasingly important in deployment contexts.

- The OOD datasets (e.g., HellaSwag, MedQA) share similar formats and reasoning structures (mostly multiple-choice) making this evaluation closer to cross-task transfer within a single distribution family rather than genuine OOD testing. True OOD robustness would require testing on unseen prompt styles, task types, or language domains, which is currently missing.

- The paper compares MARVEL primarily against merging baselines, which are not designed for abstention. It omits direct comparisons to alignment or uncertainty-aware tuning methods that explicitly model abstention or confidence. Without these baselines, the claimed improvements remain incomplete.

- The interpretability analysis (Figs. 4–6) is descriptive rather than causal. While activation histograms suggest expert specialization, the router is not trained on rationale labels, so these correlations may arise spuriously or reflect LoRA scaling differences. The paper currently misses controlled probing to validate these claims.

**Questions:**

See above

---

### Note · Authors · 2025-12-16

I have read and agree with the venue's withdrawal policy on behalf of myself and my co-authors.